# A quantitative microbial risk assessment for touchscreen user interfaces using an asymmetric transfer gradient transmission mode

**Andrew Di Battista** *

Ultraleap Ltd., Bristol, United Kingdom

* andrew.di.battista@ultraleap.com

## Abstract

The ubiquitous use of public touchscreen user interfaces for commercial applications has created a credible risk for fomite-mediated disease transmission. This paper presents results from a stochastic simulation designed to assess this risk. The model incorporates a queueing network to simulate people flow and touchscreen interactions. It also describes an updated model for microbial transmission using an asymmetric gradient transfer assumption that incorporates literature reviewed empirical data concerning touch-transfer efficiency between fingers and surfaces. In addition to natural decay/die-off, pathogens are removed from the system by simulated cleaning / disinfection and personal-touching rates (e.g. face, dermal, hair and clothing). The dose response is implemented with an exponential moving average filter to model the temporal dynamics of exposure. Public touchscreens were shown to pose a considerable infection risk ($\sim 3\%$) using plausible default simulation parameters. Sensitivity of key model parameters, including the rate of surface disinfection is examined and discussed. A distinctive and important advancement of this simulation was its ability to distinguish between infection risk from a primary contaminated source and that due to the re-deposition of pathogens onto secondary, initially uncontaminated touchscreens from sequential use. The simulator is easily configurable and readily adapted to more general fomite-mediated transmission modelling and may provide a valuable framework for future research.

## Introduction

The prevalence of shared Touchscreen User Interfaces (TUIs) has become apparent in recent years; whether it be a fast-food menu or an airport terminal self check-in machine. However, their reputation for hygiene has come under scrutiny, predominantly from sensationalised media articles [1–3]. The fact that touchscreens carry pathogens is not however in question here; what is yet to be established is if they can transmit enough pathogens to a user so as to cause infection (and if so, which disease?). Thus, the aim of this work was to carry out a

**Data Availability Statement:** The code is freely available at: https://github.com/andydiba/fomite_sim2. Contact the author for any questions.

**Funding:** ADB is an employee of Ultraleap Ltd. As sponsors, Ultraleap played no role in the study

design, data collection and analysis or preparation of the manuscript. The intention to publish this work and to provide open-source data/code, was arranged with Ultraleap prior to the start of this project. Ultraleap has funded the publication costs of this paper. The funder provided support in the form of salaries for authors ADB, but did not have any additional role in the study design, data collection and analysis, decision to publish, or preparation of the manuscript. The specific roles of these authors are articulated in the 'author contributions' section.

**Competing interests:** I have read the journal's policy and the authors of this manuscript have the following competing interests: his was a collaborative study funded by Ultraleap Ltd. Ultraleap is a manufacturer and developer of human-computer interface technology. ADB is a paid employee of Ultraleap. Ultraleap's role was limited to funding these costs as well as any fees associated with publication. This does not alter our adherence to PLOS ONE policies on sharing data and materials.

Quantitative Microbial Risk Assessment (QMRA) using computer simulation methods, in an attempt to answer these questions [4].

## Surface contamination

Studies examining the typical microbiome of public TUIs have shown that it is made up of largely innocuous microbes found also on the human hand [5]. However, harmful pathogens such as *Enterococcus*, *E.coli* and *Staphylococcus aureus* have also been found in concentrations typically $\sim 1$ *CFU*/$cm^2$ [6, 7] but as high as 60 *CFU*/$cm^2$ [8]. These concentrations may be taken as a baseline contamination level.

Higher levels of contamination can occur when considering droplet deposition from someone suffering from a respiratory illness such as influenza. Viral concentrations in these droplets are on the order of $\sim 10^4$ *PFU*/$ml$ [9, 10]. The average volume of a cough being between 0.006—0.044 ml (x 40 for a sneeze) [11–13].

Unwashed hands, particularly after toilet use, is another concerning source of contamination. It has been estimated that 30% of individuals do not wash their hands sufficiently [14–16] and there are additional issues in lavatories with regards to using contaminated soap [17] and doorknobs [18]. Bacterial load found on hands after toilet use ranges from $0.85 \pm 0.93(SD) \times 10^5$CFU for washed and dried hands to $3.64 \pm 4.49(SD) \times 10^5$ CFU for unwashed hands [19]. In other experiments, $10^8$ CFU/g has been used to approximate natural bacterial contamination levels [20]. Thus, a TUI with heavy use (and little cleaning) may accumulate a considerable concentration of pathogens. This is exacerbated by the fact that traditional TUI design would have each user touching the same locations on the screen e.g. an *OK* or *Pay Now* button, concentrating the effective area of the screen irrespective of the actual physical dimensions.

## Transfer coefficients

Transfer coefficients (or rates) are used to describe the proportion of microbes transferred from a fingertip to a surface (and vice versa) after a touch event. There is a wealth of experimental evidence to show that an *asymmetry* exists between transfer coefficients dependant on which (finger or surface) is the source or destination of initial microbial contamination [21–23]. Moreover, they are dependant on the touch pressure, duration, surface porosity and environmental conditions such as humidity and temperature. Thus, transfer coefficients may be better described through a stochastic process.

More recent experiments have demonstrated that transfer coefficients also depend on the *difference* between microbial concentration on the finger and the surface being touched [24, 25]. This *gradient* assumption, in the absence of pathogen die-off or other sources of loss, leads to a *dynamic equilibrium* being reached after multiple touch events. Essentially, just as many microbes from the finger are *deposited* onto a surface as are *picked up* from the surface.

A contribution of this paper is combining both asymmetry and gradient properties of microbe transfer into a single model (Transfer Rates: Asymmetric Gradient Model). This model also gives the ability to simulate the re-deposition of microbes (i.e. cross-contamination of surfaces) and assess the relative infection risks of direct and secondary exposure.

## Effectiveness of cleaning & disinfection

Given the uniformly smooth non-porous surface of a TUI one might assume that traditional wiping with a cleaning and disinfectant agents would prove highly effective at controlling surface microbes.

Traditional chemical based sprays and wipes have been shown to be up to 98% effective at removing pathogens [26]. Alternatives to chemicals are UV light [27–29] and the emerging

technology of self-cleaning antimicrobial surface coatings, many of which are commercially available e.g. for tablets/smart-phones [30–32].

The effectiveness of all of these methods is reduced when administered inappropriately or by insufficiently trained staff. Also, they may not be suitable for controlling all pathogen types. Any cleaning regimen requires ongoing monitoring and measurement of bioburden to truly ascertain its effectiveness [33]. In a network of people and public TUI use, timing is a key factor that is explored and discussed in Fig 7 and Decontamination.

### Dose response

The traditional paradigm of fomite-mediated transmission assumes that a person will at some point touch their face ($\sim$ 20 times per hour) and transfer pathogens from their fingers to their mucosal membranes (mouth, nasal passages and/or eyes) i.e. *self*-inoculate [34, 35]. Mucus plays an important role in the immune response and can also be a source for pathogen spread [36, 37]. The period of time over which an individual is inoculated has a significant impact on infection risk. This *dynamic dose response* is explored in [38, 39]. Essentially, it is assumed that pathogens experience a form of exponential decay once inside a mucus membrane due to natural die-off / inactivation in conjunction with immune system activity. This paper uses a dynamic dose response while accounting for the sporadic influx of pathogens from TUI use and self-inoculation (Dose Response).

### Previous work

In a initial attempt at a QMRA for touch screens [40], a complex network of TUI interactions was simulated. It suggested that parameters such as pathogen infectivity and survival time as well as the rate of surface decontamination play an important role in determining risk. One shortcoming of this work was that the model did not include a gradient transfer model and so could not model the effects of re-deposition of pathogens / cross-contamination of surfaces. Building upon this initial work, this paper aims to overcome these deficiencies.

### Summary

The remainder of this paper will describe a stochastic simulation model with particular attention to the calculation of microbial transfer (Transfer Rates: Asymmetric Gradient Model & Additional sources of pathogen removal) and dynamic dose response (Dose Response). Simulation results will be presented from a hypothetical scenario involving customers making use of two public TUIs in a commercial retail setting (Results). The risk of infection from TUI use as well as a discussion of the sensitivity of key simulation parameters are presented in Discussion.

## Methods

The core of the fomite-mediated transmission paradigm used in this paper is summarised in Fig 1. The key features (highlighted in bold) are described separately in the subsequent sections. For each parameter used in the simulation, default values are listed in Table 1 of the Appendix D in S1 Appendix.

### Queueing network

An important feature of this QMRA is its ability to simulate the flow of people and their TUI interaction. This is accomplished using a system of first-order Markov Chains and FIFO (first-in-first-out) queue structures at each zone / location. Details of the implementation are found

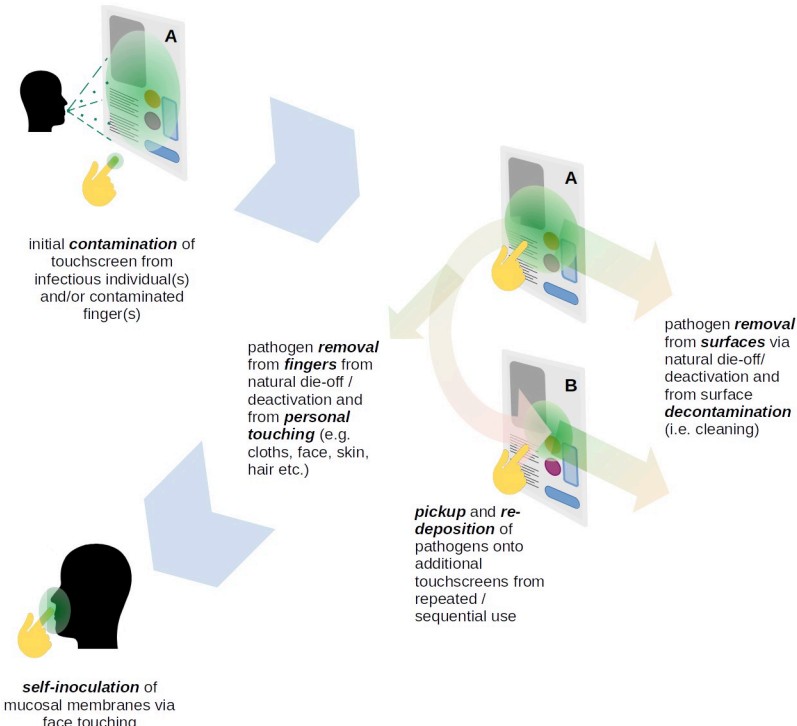

**Fig 1. Disease transmission via TUI.** A basic illustration of key model features and chain of events.

in Fig 2 and Appendix A in S1 Appendix. The reader is directed to [41] for further theoretical background.

## Transfer rates: Asymmetric gradient model

The data concerning transfer coefficient asymmetry and gradient behaviour can be combined and incorporated into a single model (Eqs 1 and 2). Let the initial concentrations (number of microbes / unit area) on a surface or finger be denoted as $C_{s,0}$ and $C_{f,0}$ respectively. For TUI

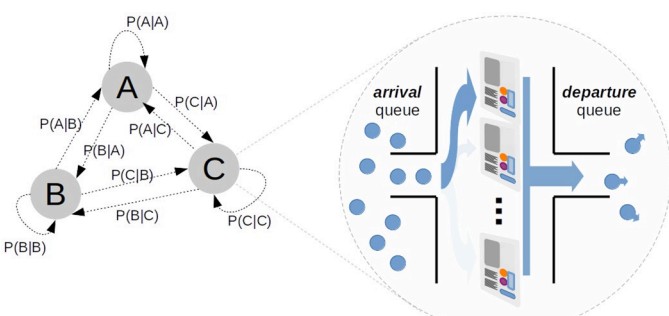

**Fig 2. Queueing Network and Markov Chains.** The flow of people from one zone to another can be controlled by a first-order Markov Chain (see. Eq A.1 in S1 Appendix) Upon arriving at a location, individuals will queue to use one (of possibly many) identical TUIs (e.g. a bank of ATM machines). After interacting with the TUI, they join a 'departure' queue (effectively hanging around at that location) until finally rejoining the Markov Chain and moving on to a new zone/location. A pair of rate parameters associated with a particular TUI type ($\sim Pois(\lambda_{TUI})$) and zone ($\sim Pois(\lambda_Z)$) determine how long they queue in the 'arrival' and 'departure' queues respectively.

interaction, it is assumed the same effective area of the screen is being contacted with the same finger. Consequently, one can reinterpret $C_s$ and $C_f$ simply as being proportional to the *number* of microbes. Further, one can *normalise* these values by the total number of microbes such that $C_s + C_f \equiv 1$. Consider a time interval in which $n$ repeated identical touch events take place; $C_s$ and $C_f$ can be updated as follows:

$$C_{f,n} = (C_{f,n-1} - (\alpha C_{f,n-1} - \beta C_{s,n-1}))d_f \tag{1}$$

$$C_{s,n} = (C_{s,n-1} + (\alpha C_{f,n-1} - \beta C_{s,n-1}))d_s \tag{2}$$

where $\alpha$ deposit rate (from finger to surface), $\beta$ is the deposit rate (from surface to finger) and $d_f$ and $d_s$ are the pathogen survival rates (% survival / unit time) on fingers and the surface, respectively. Consider the case of either a clean finger touching a contaminated surface ($C_{f,0} = 0$)) or a contaminated finger touching a clean surface ($C_{s,0} = 0$)). From these scenarios it becomes evident that $\alpha$ and $\beta$ can be estimated from empirical results of transfer asymmetry experiments already mentioned in [21–23].

It can also be shown from Eqs 1 and 2, that in *lossless* conditions (i.e. $d_f = d_s = 1$), a dynamic equilibrium is reached between finger and surface pathogen levels. The equilibrium point is found by noting:

$$\underbrace{\alpha C_{f,\infty} - \beta C_{s,\infty} = 0}_{equilibrium} \quad and \quad \underbrace{C_{f,\infty} + C_{s,\infty} = 1}_{lossless} \tag{3}$$

which results in:

$$C_{f,\infty} = \frac{\beta}{\alpha + \beta} \quad and \quad C_{c,\infty} = \frac{\alpha}{\alpha + \beta} \tag{4}$$

By Incorporating pathogen decay (i.e. $0 < d_f, d_s < 1$), it is evident that all pathogen levels will eventually go to 0 as $n \rightarrow \infty$.

This can be seen by simulating repeated touch events between a single person and a touchscreen (Fig 3). Default parameters are used throughout this simulation for $\alpha$ and $\beta$ for a finger in contact with a non-porous surface (e.g. glass, ceramic tile) [22–24]. Note: when 'averaging' the net effect of random transfer coefficients the *geometric* mean is used; similar to interest rates, transfer coefficients compound in a multiplicative sense.

## Additional sources of pathogen removal

**Loss to environment/clothing.** An additional source of pathogen loss in this system comes from an individual touching other surfaces, namely their skin, hair and clothing. This rate of *personal-touching* can be as high as 200 per hour [42]. In our simulation, all individuals are assumed healthy; so any microbes *picked up* from personal-touching would correspond only to innocuous human microbiota. For the purposes of a QMRA this can be ignored. Also, in the case of clothing the rate of transfer from fabric back to the finger is very low (<1%) [43], particularly when both surfaces are dry. When combined with the fact that any of these surfaces are unlikely to be touched repeatedly *in exactly the same spot* personal-touch events can be treated as a 'sink' for finger pathogens. Mathematically, one simply ignores Eq 2 and set $C_{s,n} \equiv 0 \; \forall n$ in Eq 1.

The deposit rate (finger to clothing) ($\alpha$ from Eq 1 that can be denoted as $\alpha_{pt}$ to distinguish it from TUI related parameters) is a parameter that is explored later in the simulation; default

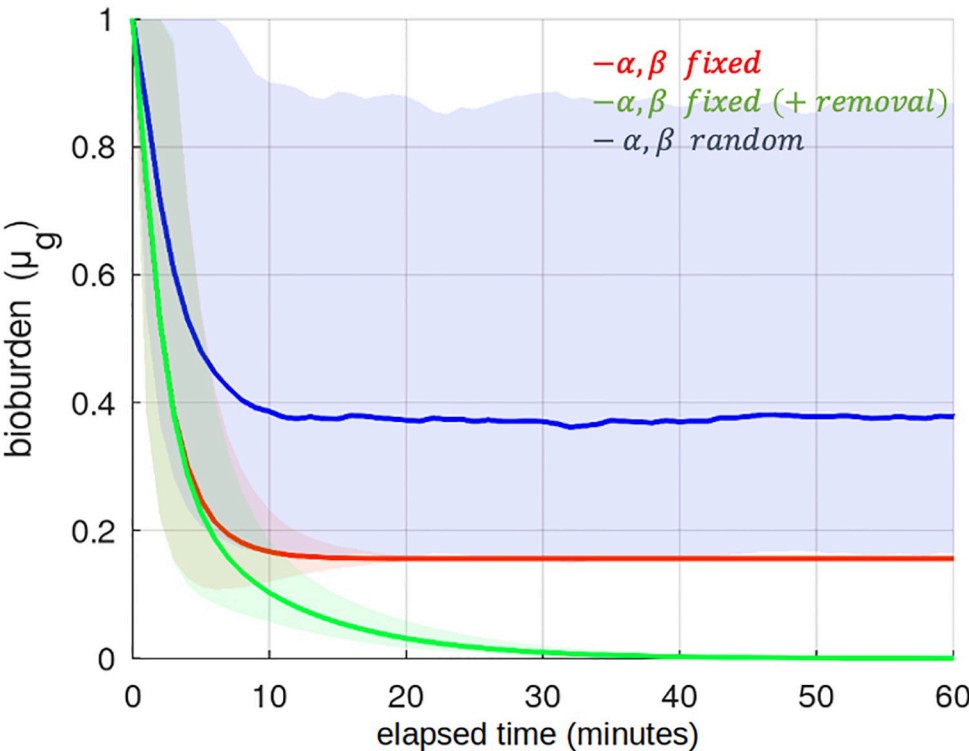

**Fig 3. Dynamic equilibrium of surface bioburden.** A single individual (with clean hands) continuously touches a surface at the same location(s) e.g. the 'buttons' on a TUI. The geometric mean and standard deviation (GSD) of surface bioburden is depicted with three variations of the pickup/deposit rate model: fixed parameters ($\alpha = 0.05$, $\beta = 0.27$) with (green) and without (red) loss due to pathogen decay/die-off, and (blue) random parameters $\alpha \sim f_{tn}(0.05, 0.3, 0, 1)$ (median average 0.22) and $\beta \sim f_{tn}(0.27, 0.3, 0, 1)$ (median average 0.34) where $f_{tn}(\mu, \sigma, a, b)$ is a truncated normal distribution on the domain $[a, b]$. The plots are consistent with Eq 4, which predicts an equilibrium point of 0.15 for the fixed parameter case and $\approx 0.40$ for random parameters. Once pathogen loss is incorporated, all levels tend to 0 over time.

values are listed Table 1 in S1 Appendix. With regards to face touching events, it is assumed to be a random transfer coefficient, $\sim f_{tn}(0.35, 0.1, 0, 0.6)$ [24, 35].

**Cleaning efficiency.** Based on chemical wipes, cleaning effectiveness is modelled using a log-normal distribution with modal value of 94% [44]. It should be noted that this is likely an overestimate as typical alcohol based gels may be hindered by other factors such as the presence of mucus (with survival exceeding 10%) [45]. Cross-contamination from improper cleaning methods is not modelled. In the context of this simulation, chemical cleaning and alternatives like UV light are largely comparable in terms of effectiveness. It is more the *rate* at which cleaning is carried out on TUIs that is a parameter of interest for this simulation. On the other hand, anti-microbial surface coverings have different characteristics that will be discussed in Decontamination.

## Dose response

The human $ID_{50}$ is defined as the infective dose with 50% probability of infection. For example, respiratory viruses typically require a relatively large dose for infection ($10^3 – 10^4$ PFU) [11, 46, 47]. For many types of bacteria and enteric viruses this can be as low as 10—100 CFU (or PFU) respectively [48, 49]. If one were to interpret these values as estimates for $ID_{50}$, provided one sticks with the same units, pathogen levels can effectively be normalised and made into a generic type for the purposes of mathematical simulation.

It is customary to model a dose response using an exponential cumulative distribution function (CDF) [12]. The probability of infection, $P$, at any time, $t$, will depend on the effective *accumulated* pathogen load in an individuals mucus membranes, $D(t)$.

$$P(t) = 1 - e^{-\ln(2)\; D(t)/ID_{50}} \tag{5}$$

$D(t)$ will vary continuously as an individual interacts with TUIs and self-inoculates. Let each self-inoculation event at time $t_i$ result in an instantaneous dose $d_{t_i}$. In order to capture the dynamic characteristics of this exposure, an exponential moving average filter characterised with a time-constant parameter $\gamma$ is used.

$$D(t_i) = d_{t_i} + \left(1 - \frac{\Delta t_{step}}{\gamma}\right) D(t_{i-1}) \tag{6}$$

where $\Delta t_{step}$ is the time-step (resolution) of the simulation e.g. 1 minute ($\gamma$ should be defined using the same units). It should be clear that Eq 6 implies an exponential decay (die-off) of pathogen in mucus. $\gamma$ can be referred to as the *inoculation period* as it describes a time-interval over which pathogens can contribute to infection risk. An illustration of the effects of different values for $\gamma$ can be found in Appendix C in (Fig 8) in S1 Appendix.

## Simulation

For the main simulation the aim is to provide a simple yet plausible scenario that would allow for closer examination of the effects of each governing parameter. Moreover, in order to examine the effects of re-deposition of pathogens i.e. cross-contamination it would seem essential to include at least two TUIs.

One example of sequential TUI use would be airport self-checkin and bag-drop machines. A more generic case (used in this simulation) is in retail outlet centres (shopping malls) where customers may venture from one shop to another making use of a TUI to browse and complete transactions (Fig 4).

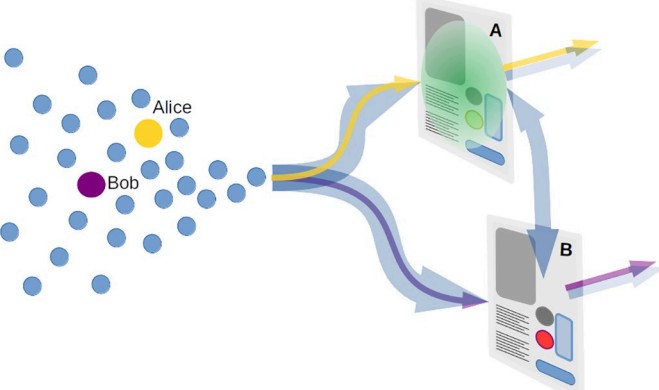

**Fig 4. Simulated scenario.** Consider two shops (A & B) with customers (n = 32) arriving at each randomly, with equal probability. Some customers (2/3) at A will move on to B afterwards (and vice-versa). The remaining 1/3 will visit only one of the shops (it can be assumed that they then move on to other locations outside the simulation's focus). The TUI at shop A has already been contaminated with an initial bioburden. There is a particular interest in Alice and Bob; Alice will visit shop A exclusively while Bob will visit shop B exclusively. Therefore, Bob can only be infected from secondary transfer of pathogens from shop A (even though he will never visit A). Alice will receive 'first-hand' exposure to the contaminated screen. How does the risk of infection differ for the these two individuals?.

Everyone enters the simualtion from an initial population pool at an average rate $\sim Pois$ (0.25) i.e. once every 4 minutes. Thus, the simulation was expected to take approximately 2 hours. Alice and Bob are programmed to enter the simualtion $\sim Pois$(0.0167) i.e. once per hour; they arrive as if from the middle of the initial population queue (on average).

**Touch rates.** As with transfer rates, the number of touches, $n_t$, expected to complete a transaction or menu selection can be modelled using a truncated normal distribution. For example, an ATM pin pad requires a minimum of 5 touches (4 pin numbers + OK), i.e. both mode and minimum can be set to 5. However, cancelled transactions, re-attempts, correcting invalid input etc. means that one can expect a small variance in touch numbers. The simulation should ideally account for additional touches associated with (for example) browsing for products. The default simulation parameters $f_{tn}(\mu, \sigma, a, b) \rightarrow f_{tn}(12, 3.5, 8, 40)$ incorporate this random behaviour.

Another important assumption in this simulation is that people will interact with a TUI using just one finger e.g. index finger of the dominant hand. As a consequence, a somewhat crude simplification is to scale down by a factor of 10 all touch rates associated with face/ personal-touching quoted in the literature.

## Simulation code

The simulator was implemented in the C programming language. Custom configuration files were used to set and load simulation parameters. Simulation outputs were exported to.csv files, ready for plotting and further analysis. The code is available as open-source at https:// gitlab.com/fomite-simulator/fomite_sim_v2. A *readme* file provides an introduction for compilation and use, along with sample configuration files.

## Results

Each simulation result is averaged over N = 40000 realisations in order to achieve a suitable level of confidence in the output (Appendix B in S1 Appendix). The simulation had a time-step (resolution) of 1 minute. In all figures, the light orange curve is associated with the infection risk exclusive to location **A** (i.e. infection risk to Alice); infection from the primary source of contamination. The purple curve is associated with second-hand exposure at location **B** (i.e. infection risk to Bob).

## Surface cross-contamination

In Fig 5 the (geometric) mean average bioburden on TUI A and B is plotted over time.

## Simulation parameter sensitivity

Presented here are the effects of several key model parameters on infection risk.

Fig 6(a) depicts the effect of initial bioburden on **A** over a range of $ID_{50}$. Fig 6(b) and 6(c) show the infection risk associated pathogen decay (half-life) and inoculation period ($\gamma$), respectively. Note: pathogen half-life, $l_{1/2}$, is related to parameter $d_s$ from Eq 1 by $t_{1/2} = -\ln2/\ln d_s$.

Fig 6(d) depicts the infection risk as a function of the average number of touches required for TUI interaction. Fig 6(e) illustrates the effect of increasing the number of available touchscreens for use at B. In order to establish the importance of pathogen loss due to personal-touching events, various deposit rate coefficients are considered Fig 6(f).

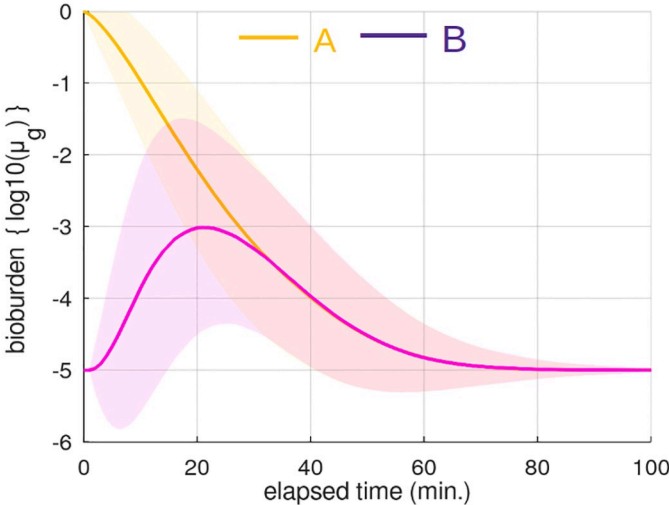

**Fig 5. Bioburden.** Geometric mean bioburden (log10 scale and normalised) and standard deviation (GSD) on TUIs **A** and **B**. **A** is initially contaminated. Over time microbes are transferred to **B** (0.1% avg. at 20 minutes) through sequential use. The eventual decay of both curves can be explained by pathogen die-off, personal-touching and the fact that 1/3 of people will visit only one location before leaving the simulation (mitigating further cross-contamination between **A** and **B**).

It can be seen from Fig 6 that infection rates for Alice is approximately 2.5% to 3% at default parameter settings (see Appendix D in S1 Appendix). The risk to Bob is typically half that of Alice.

## Effects of surface cleaning

Fig 7 show the effects of increasing the rate of cleaning/disinfection at each location and the associated risks to A and B.

## Discussion

### Suitability of simulated scenario

The rationale behind the simulation scenario was to provide a risk assessment of TUI use from a generic pathogen type under plausible circumstances. Comparing risk from primary and second-hand exposure was an important and distinctive property of the asymmetric gradient transmission model. The model allowed for estimation of an individual's finger bioburden over time and subsequent cross-contamination of surfaces. Two individuals (Alice and Bob) were used to represent the personal risk of using a TUI at two locations (**A** and **B**) respectively. By having a proportion of the population move on after one visit/interaction a wider network of shops and TUIs was effectively incorporated into the model; the effect was a more rapid removal of pathogens from the overall system.

As a simplifying assumption, the total bioburden in each simulation was fixed. Another option would be to include additional time-dependant sources such as infectious individuals coughing or sneezing in the vicinity of a TUI. It is reasonable to deduce that an introduction of more pathogens would lead to higher infection rates than those presented in the results of this simulation.

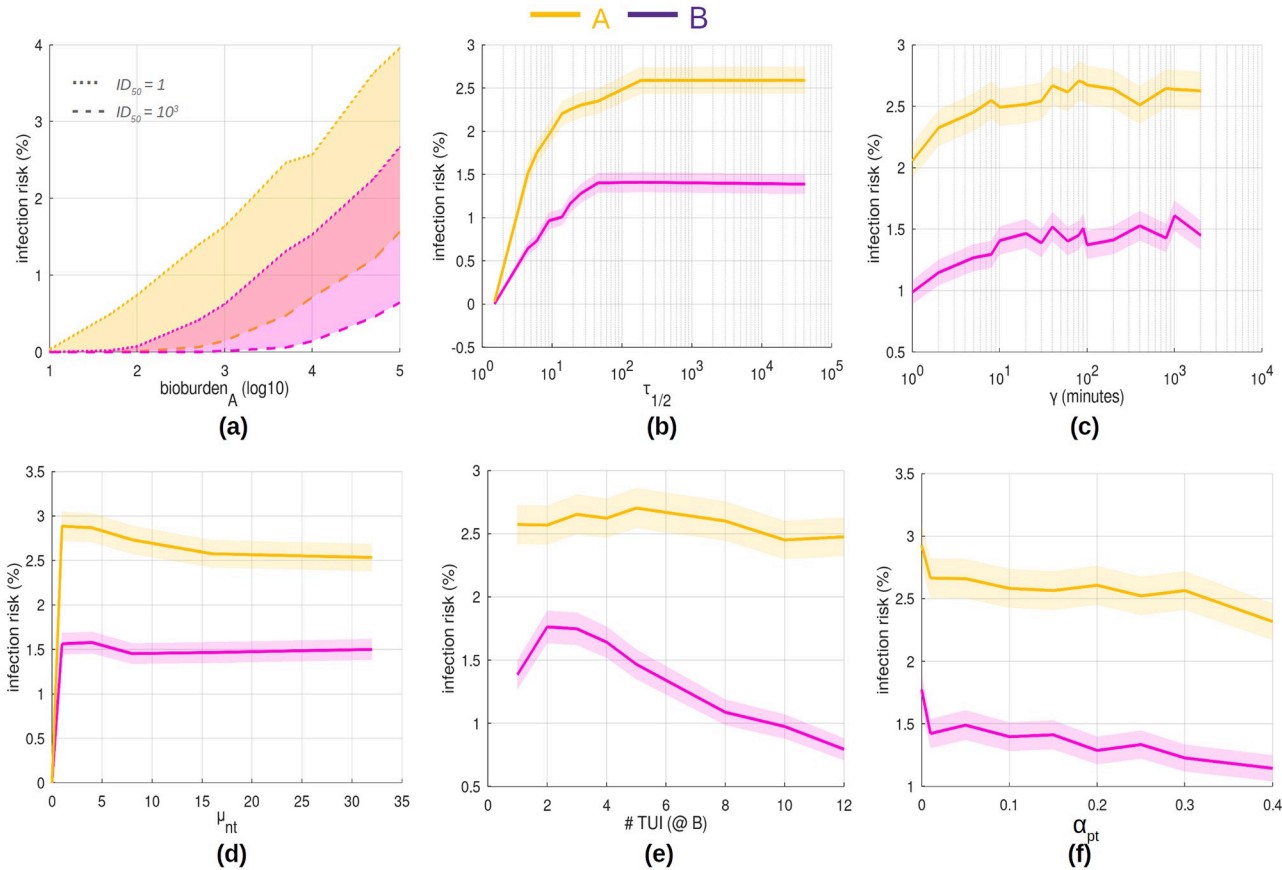

**Fig 6. Parameter sensitivity on infection risks.** Average infection risk (n = 40000 realisations) with 95% CI. (a) Initial bioburden seeded on TUI **A** (over a range of $ID_{50}$) (95% CI omitted for clarity). Bioburdens range from baseline ($\sim 10^1$) to 'heavy' contamination ($10^5$). (b) Pathogen surface half-life (minutes). (c) Inoculation period (time-constant $\gamma$ of the dynamic dose response model). (d) Average number of touch interactions, $n_t$ required for TUI use. (e) Number of TUIs available for use at **B**. (f) Deposit rate associated with personal-touching events $\alpha_{pt}$.

## Contamination levels

It is clear from Fig 6(a) that in order to reach a significant infection risk to Alice or Bob the ratio of pathogen contamination number (bioburden) to $ID_{50}$ needs to be large. When considering baseline levels that have been associated with public TUIs ($\sim 1\ CFU/cm^2$), a pathogen would need to be highly infectious to be viable for disease transmission. More significant risk is associated with higher initial contaminations levels; a result of possibly poor hygiene practices from customers and/or cleaning regimens at the establishment.

## Dose response

From Fig 6(c) it appears that the risk associated with an increasing inoculation period ($\gamma$) begins to plateau. This may be largely due to the total bioburden being fixed; individuals will reach some expected maximum limit of pathogen numbers in which they can self-inoculate with. When considering a strong innate immune response (associated with ever decreasing $\gamma$) infection risk is reduced by $\sim 30\%$ for both Alice and Bob.

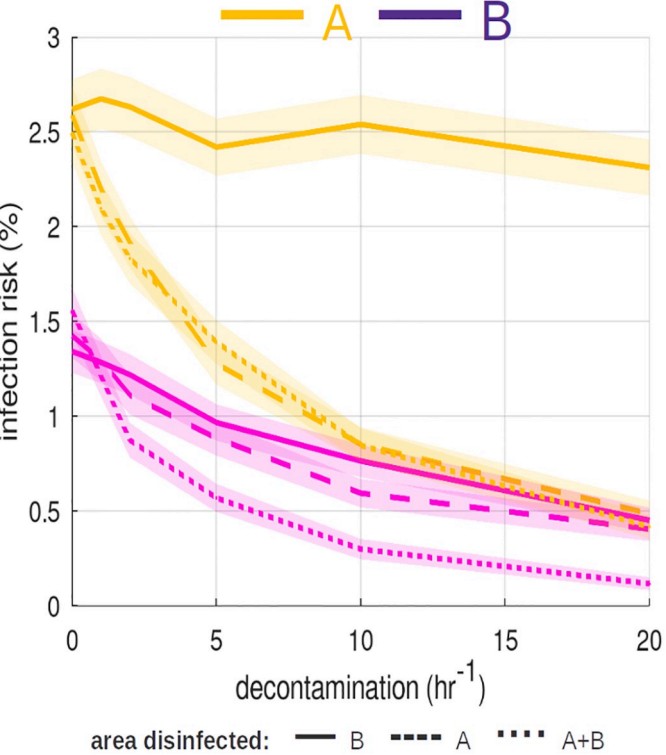

**Fig 7. Cleaning/disinfection rates.** Average Infection risk (n = 40000 realisations) with 95% CI. Solid lines depict infection risk when cleaning rates applied at location **B** only. Dashed lines are associated with cleaning at **A** only. Dotted lines correspond to cleaning rates applied both at **A** and **B**. Decontamination at **B** alone has little impact on the risk to Alice (though a slight decrease is observed at higher cleaning rates as pathogens are prevented coming full-circle back to **A** from **B**). Decontamination at **A** alone has slightly better outcomes for Bob than cleaning at **B** alone. This suggests the risk to customers at one location can be subject to hygiene standards and cleaning regimens at *another independent* location.

## TUI design

As the number of touches required to complete a menu selection or transaction increases one would intuitively expect the infection risk to increase (Fig 6(d)). However, this is not the case. There are several factors to consider: (1) the gradient transmission model ultimately limits bioburdens on fingers and surfaces via dynamic equilibrium (2) pathogens are removed through die-off *but also* personal-touching and self-inoculation events. The more pathogens picked up off surfaces, the more are removed from the system altogether. This has a significant impact namely because of (3) fixed total initial bioburden on **A**. Essentially, both Alice and Bob were shielded from further risk because everyone else was carrying pathogens away with them.

Having more TUIs to choose from at **B** meant a *de-concentration* of surface bioburdens and a reduced infection risk for Bob (Fig 6(e)). A rather perplexing exception is the slight rise in risk for 2 to 4 TUIs. One explanation could be related to the flow of people: both Alice and Bob were programmed to arrive at their destination such that they were (on average) in the middle of the population queue of 32 people (based on 1 TUI per location). By altering the number of TUIs the throughput at **B** is changed; Bob arrives sooner (possibly when pathogen deposits are 'fresher'). After a certain threshold, the de-concentration effect takes over. This highlights the complexities that may arise within queued networks.

An interesting approach to hygienic TUI design suggests to shift the layout of the interface as it appears on screen, thus removing the constraint of forcing individuals to touch the exact same area [50]. This would effectively create multiple *virtual* TUIs to choose from as in Fig 6 (e) (without actually increasing throughput).

## Personal-touching

In Fig 6(f) the average deposit rate coefficient associated with personal-touch events was altered. Though not dramatically so, higher rates do lead to reduced risk as more pathogens are transferred to clothing etc. and removed from the system. The importance of personal-touch events in modelling fomite-mediated transmission is apparent but remains a complex behaviour to completely replicate.

## Decontamination

Throughout the simulation it was assumed that individuals did not *deliberately* wipe/wash their hands between TUI. Such habits would, of course, drastically reduce infection risk associated with fomites. However, enforcing stringent hand-washing policies is not an attractive option for retail businesses and may not be feasible in all circumstances.

An alternative approach that places the responsibility and control with the TUI owners/operators is enhanced sterilization regimens. From Fig 7 it is apparent that cleaning can have a significant impact on infection risk. However, if the goal is to virtually eliminate the risk ($\ll$ 1%), disinfection would have to take place every few minutes to keep up with constant cross-contamination from other sources.

An interesting result in Fig 7 is from comparing Bob's infection risk when cleaning is carried out at **A** exclusively and **B** exclusively. The infection risk is lower when the *original source* of contamination (i.e. **A**) is cleaned. This suggests the risk to customers at one location can be subject to hygiene standards and cleaning regimens at *another independent* location. Ultimately, the infection risk to Alice was consistently higher than that of Bob suggesting that second-hand exposure can only succeed that of the primary source of contamination.

Anti-microbial coatings are designed to reduce pathogen survival rates on surfaces. For this reason, Fig 6(b) best illustrates their possible benefits. It is clear that a well-designed coating can have significant impact on infection risk; driving it virtually to 0 in the extreme case. However, further developments still need to be made to meet this performance level across a sufficiently wide range of pathogens types [33, 51].

Touchscreens can be designed or retrofitted to employ automated cleaning cycles using UV light. An automated cycle after each use would greatly reduce surface bioburdens and infection risk. However, there would be an associated added cost for the equipment and increased energy consumption. Dependant on power output, the UV cycle may take several seconds to operate which could adversely affect user throughput. Additionally, the effectiveness of UV light is hindered by dirt and other fluids; therefore TUIs would still require a traditional cleaning. UV itself may also poses potential health risks (though an automated system would least reduce exposure risks to staff employed to clean) [52]. Yet another risk posed by decontamination methods, including UV, is the inadvertent selection for microbial resistance [53].

An alternative to decontamination altogether is to re-develop or retrofit existing TUIs with hand tracking cameras to provide a touch-less interface. This nullifies surface microbial transfer and eliminates infection risk (Fig 6(d)). Whether or not businesses and venues choose to implement such alternatives will likely depend on the cost of cleaning, replacement or conversion of existing TUIs.

## Risk assessment

In these simulations, infection occurred (for Alice) as much as $\sim 3\%$ of the time. It is worth discussing whether or not this poses a significant risk. Firstly, infection *risk* in an epidemiological context really refers to the *likelihood* or probability of infection. Risk, in a heath and safety context, is often defined as a product of the *likelihood* and the *severity* of an adverse event. While this simulation was capable of computing the former, there was no function implemented to assess the cost to health, long or short term, nor any subsequent economic losses due to disease spread.

For comparison, another study to asses the role of fomite transmission in a simulated norovirus outbreak predicted a likelihood of infection ranging widely from 5% to 57% [54]. Similar values were also found in an assessment of fomite transmission in a real world SARS outbreak [55]. However, in the case of SARS, other transmission routes e.g. airborne played a dominant role.

It seems possible then that TUIs may play a small contributing role in larger outbreaks. The reason why they do not gain more attention or blame in scenarios where touch is the dominant form of transmission may have something to do with the severity and timing of illness. Consider, in this simulation, if Alice or Bob were to eventually develop mild to moderate gastroenteritis the following evening. Would they attribute it to a single interaction with a TUI? Would their illness eventually be reported and counted by the local health authority? Without this data, it is easy to see why epidemiologists still strive to understand the nuances of fomite mediated transmission.

Pathogen survival (Fig 6(b)) and infectiousness (Fig 6(a)) are key factors in determining which pathogens are likely to be responsible for TUI disease transmission. It is worth noting that bacteria have the capacity, theoretically, to survive and even grow on surfaces given the right conditions. Viruses, on the other hand, can only diminish. It is interesting to note in Fig 6(b) that infection rates level-off when pathogen half-life exceeds $\sim 100$ minutes; after which pathogen removal is dominated by personal-touching (Fig 5). Therefore, when considering pathogen survival it should be brought into the context of TUIs; it is not necessary for a pathogen to survive long, merely *long enough* given the rate of TUI use.

The default survival rates used in this simulation where derived from available laboratory reports of a select group of pathogens on glass/ceramic surfaces. Data from actual TUIs could reveal different results and could, for example, take into account any effects from the light / heat emanating from the screen. Moreover, techniques for *quantifying* bacteria and viral particles is prone to wide margins of error. Therefore, the simulation is only as reliable as the data put into it.

## Model limitations

An important limitation of the model was to assume all interaction took place with a single dominant finger. A further consequent simplification was that the area on the TUI was effectively homogenized. This allowed for the transfer coefficients to be modelled as simple individual random variables (see Appendix D in S1 Appendix).

In order to model more complex interactions using multiple fingers, it would be necessary to consider not only an extended set of transfer coefficients but also partition the TUI surface into regions where each finger is likely to touch. There could also be cross-contamination between an individual's fingers. Essentially, Eqs 1 and 2 would need to be implemented for each finger and for each possible touch interaction. This would add a significant level of complexity to the model but may be useful in assessing risk associated with a specific TUI design/

layout. The advantage of the current simplified model is that it offers generalised insights into how TUI and networks of people interact to spread disease.

## Simulator framework

The implementation of the simulator was developed in C which offers good performance for computationally intensive operations (see Code availability). The simulator can be operated using a (relatively) easy-to-use configuration file system, making it straightforward for a researcher to simulate complex queueing networks and TUI types without re-compiling code. It is hoped that this may provide a valuable framework for future research endeavours in fomite-mediated disease transmission modelling.

## Supporting information

**S1 Appendix.**
(PDF)

## Acknowledgments

The author would like to acknowledge the support of Orestis Georgiou for his insights and encouragement in the realisation of this research project.

## Author Contributions

**Conceptualization:** Andrew Di Battista.

**Data curation:** Andrew Di Battista.

**Investigation:** Andrew Di Battista.

**Methodology:** Andrew Di Battista.

**Software:** Andrew Di Battista.

**Validation:** Andrew Di Battista.

**Writing – original draft:** Andrew Di Battista.

**Writing – review & editing:** Andrew Di Battista.

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
