## [Decision Letter · Decision Letter 0]

26 Nov 2021

PONE-D-21-23569

A quantitative microbial risk assessment for touchscreen user interfaces using an asymmetric transfer gradient transmission mode

PLOS ONE

Dear Dr. DiBattista,

Thank you for submitting your manuscript to PLOS ONE. After careful consideration, we feel that it has merit but does not fully meet PLOS ONE’s publication criteria as it currently stands. Therefore, we invite you to submit a revised version of the manuscript that addresses the points raised during the review process.

Please see below the comments and suggested MAJOR revisions made by the individual(s) who reviewed your manuscript.  If provided, the referee's report(s) indicate the revisions that need to be made before it can be accepted for publication.

We look forward to receiving your revised manuscript.

Kind regards,

Ricardo Santos

Academic Editor

PLOS ONE

“ADB is an employee of Ultraleap Ltd. As sponsors, Ultraleap played no role in the study design, data collection and analysis or preparation of the manuscript. The intention to publish this work and to provide open-source data/code, was arranged with Ultraleap prior to the start of this project. Ultraleap has funded the publication costs of this paper.”

We note that one or more of the authors is affiliated with the funding organization, indicating the funder may have had some role in the design, data collection, analysis or preparation of your manuscript for publication; in other words, the funder played an indirect role through the participation of the co-authors. If the funding organization did not play a role in the study design, data collection and analysis, decision to publish, or preparation of the manuscript and only provided financial support in the form of authors' salaries and/or research materials, please do the following:

a. Review your statements relating to the author contributions, and ensure you have specifically and accurately indicated the role(s) that these authors had in your study. These amendments should be made in the online form.

b. Confirm in your cover letter that you agree with the following statement, and we will change the online submission form on your behalf:

“The funder provided support in the form of salaries for authors [insert relevant initials], but did not have any additional role in the study design, data collection and analysis, decision to publish, or preparation of the manuscript. The specific roles of these authors are articulated in the ‘author contributions’ section.”"

“The author would like to acknowledge the support of Ultraleap Ltd. and Orestis 329

Georgiou for the realisation of this research project.”

“ADB is an employee of Ultraleap Ltd. As sponsors, Ultraleap played no role in the study design, data collection and analysis or preparation of the manuscript. The intention to publish this work and to provide open-source data/code, was arranged with Ultraleap prior to the start of this project. Ultraleap has funded the publication costs of this paper.”

“I have read the journal's policy and the authors of this manuscript have the following competing interests:

This was a collaborative study funded by Ultraleap Ltd. Ultraleap is a manufacturer and developer of human-computer interface technology. ADB is a paid employee of Ultraleap. Ultraleap's role was limited to funding these costs as well as any fees associated with publication.”

Reviewers' comments:

Reviewer's Responses to Questions

**Comments to the Author**

1. Is the manuscript technically sound, and do the data support the conclusions?

Reviewer #1: Partly

Reviewer #2: Yes

2. Has the statistical analysis been performed appropriately and rigorously? 

Reviewer #1: N/A

Reviewer #2: Yes

3. Have the authors made all data underlying the findings in their manuscript fully available?

Reviewer #1: No

Reviewer #2: Yes

4. Is the manuscript presented in an intelligible fashion and written in standard English?

Reviewer #1: Yes

Reviewer #2: Yes

5. Review Comments to the Author

Reviewer #1: The paper PONE-D-21-23569 entitled " A quantitative microbial risk assessment for touchscreen user interfaces using an asymmetric transfer gradient transmission mode" discusses an important issue which is the quantitative microbial risk assessment. In particular, the author wished to know if touchscreens can transmit enough pathogens to a user to cause an infection.

My major concerns regarding the study are the lack of validation of the results obtained. Although it would be difficult to determine the number of infections, it should be at least possible to determine if the results presented in Fig. 6 are close to reality or how they compare with other studies with infection risk in the order of 1-2.5%. Models are important to simulate scenarios but it should be possible to assess how close to reality are the results of each scenario.

One thing that is not clear to the reader is if the risk determined is high or not. How can 1-2.5% infection risk be considered? How can this be compared to other situations?

The author also discusses the use of disinfectants and UV light to decrease the risk of transmission. Could this favour the risk of selecting more resistant strains?

Specific comments are listed below:

- The Methods section should allow the repetition of the study by other colleagues. Information about the software and language used to make the simulations/model should be given here. The author added information about the simulator framework only at the end of the paper, but more information is required in the Methods section.

- The discussion section also contains results that should be presented in the results section (e.g. contamination levels, dose response, etc). A general discussion about what the impact of the different assumptions and choices made is lacking. The author presents the discussion in different sub-sections and states the assumptions made but does not provide what would be the result if another choice was made. With the computational model, the impact of some assumptions could be tested to provide support to the decisions made.

- A general conclusion to the questions presented as aims of the study is not provided.

- English should be checked throughout the text. Examples of phrases that should be improved follow:

+ “A more concerning source of contamination if from a user(s) with unwashed hands, particularly after toilet use.”

+ “All of these methods suffer when administered inappropriately or by insufficiently

trained staff.” (Methods do not suffer..:)

+ “Each simulation results is averaged (…)”

Furthermore, the author should not use the word “we” in the text. Usually scientific studies are written without referring to who did the work. In this particular case, only one author is listed and the use of "we" should be avoided.

- “Another important assumption in this simulation is that people will interact with a TUI using just one finger e.g. index finger of the dominant hand.” – This may be true for older people but younger people use more fingers in touchscreens. In the discussion, decisions such as this should be discussed and their impact assessed.

Reviewer #2: Specific comments:

1. The authors conducted an excellent study and provided an appropriate amount of literature review on the topic.

2. The parameters tested in the QMRA i.e., queueing network, transfer rates-asymmetric gradient model, dynamic equilibrium of surface bioburden and the additional sources of pathogen removal provides a thorough approach validating the simulation.

3. Introducing the coding language C in the methods section and taking a step-by-step approach of explaining how the simulation was performed would be helpful for readers to utilize the simulation model. Excellent idea to provide the link to the code used in the study and the appendix information.

6. PLOS authors have the option to publish the peer review history of their article (what does this mean?). If published, this will include your full peer review and any attached files.

Reviewer #1: No

Reviewer #2: No

---

## [Author Response · Author response to Decision Letter 0]

7 Jan 2022

Please see Response to Reviewers document submitted.

---

## [Decision Letter · Decision Letter 1]

4 Mar 2022

A quantitative microbial risk assessment for touchscreen user interfaces using an asymmetric transfer gradient transmission mode

PONE-D-21-23569R1

Dear Dr. DiBattista,

We’re pleased to inform you that your manuscript has been judged scientifically suitable for publication and will be formally accepted for publication once it meets all outstanding technical requirements.

Kind regards,

Ricardo Santos

Academic Editor

PLOS ONE

Additional Editor Comments (optional):

Reviewers' comments:

Reviewer's Responses to Questions

**Comments to the Author**

1. If the authors have adequately addressed your comments raised in a previous round of review and you feel that this manuscript is now acceptable for publication, you may indicate that here to bypass the “Comments to the Author” section, enter your conflict of interest statement in the “Confidential to Editor” section, and submit your "Accept" recommendation.

Reviewer #1: All comments have been addressed

2. Is the manuscript technically sound, and do the data support the conclusions?

Reviewer #1: (No Response)

3. Has the statistical analysis been performed appropriately and rigorously? 

Reviewer #1: (No Response)

4. Have the authors made all data underlying the findings in their manuscript fully available?

Reviewer #1: (No Response)

5. Is the manuscript presented in an intelligible fashion and written in standard English?

Reviewer #1: (No Response)

6. Review Comments to the Author

Reviewer #1: The author of manuscript PONE-D-21-23569R1 improved significantly the manuscript following the reviewers’ comments. He answered satisfactorily all questions I raised. In particular, he added a link to the simulation code and added sections discussing risk assessment and model limitations, which were may main concerns I had when reviewing the original submission.

7. PLOS authors have the option to publish the peer review history of their article (what does this mean?). If published, this will include your full peer review and any attached files.

Reviewer #1: No

---

## [Editor Report · Acceptance letter]

16 Mar 2022

PONE-D-21-23569R1 

A quantitative microbial risk assessment for touchscreen user interfaces using an asymmetric transfer gradient transmission mode 

Dear Dr. Di Battista:

I'm pleased to inform you that your manuscript has been deemed suitable for publication in PLOS ONE. Congratulations! Your manuscript is now with our production department. 

Kind regards, 

on behalf of

Dr. Ricardo Santos 

Academic Editor

PLOS ONE